# Recent Advance of Strontium Functionalized in Biomaterials for Bone Regeneration

**DOI:** 10.3390/bioengineering10040414

**Published:** 2023-03-26

**Authors:** Xin Liu, Huagui Huang, Jing Zhang, Tianze Sun, Wentao Zhang, Zhonghai Li

**Affiliations:** 1Department of Orthopedics, First Affiliated Hospital of Dalian Medical University, Dalian 116011, China; 2Key Laboratory of Molecular Mechanism for Repair and Remodeling of Orthopedic Diseases, Dalian 116600, China

**Keywords:** strontium, biomedical materials, inflammatory microenvironment, osteoblasts, osteoclasts, bioactive glasses, bioactive ceramics, metal-based materials

## Abstract

Bone defect disease causes damage to people’s lives and property, and how to effectively promote bone regeneration is still a big clinical challenge. Most of the current repair methods focus on filling the defects, which has a poor effect on bone regeneration. Therefore, how to effectively promote bone regeneration while repairing the defects at the same time has become a challenge for clinicians and researchers. Strontium (Sr) is a trace element required by the human body, which mainly exists in human bones. Due to its unique dual properties of promoting the proliferation and differentiation of osteoblasts and inhibiting osteoclast activity, it has attracted extensive research on bone defect repair in recent years. With the deep development of research, the mechanisms of Sr in the process of bone regeneration in the human body have been clarified, and the effects of Sr on osteoblasts, osteoclasts, mesenchymal stem cells (MSCs), and the inflammatory microenvironment in the process of bone regeneration have been widely recognized. Based on the development of technology such as bioengineering, it is possible that Sr can be better loaded onto biomaterials. Even though the clinical application of Sr is currently limited and relevant clinical research still needs to be developed, Sr-composited bone tissue engineering biomaterials have achieved satisfactory results in vitro and in vivo studies. The Sr compound together with biomaterials to promote bone regeneration will be a development direction in the future. This review will present a brief overview of the relevant mechanisms of Sr in the process of bone regeneration and the related latest studies of Sr combined with biomaterials. The aim of this paper is to highlight the potential prospects of Sr functionalized in biomaterials.

## 1. Introduction

Bone tissue is an important anatomical structure in the body. It has the function of protecting organs and maintaining body shape. With the development of society and the aging of the population, the demands for bone tissue reconstruction have increased. Although bone tissue has some capacity for self-remodeling, large areas of bone defects are difficult to self-heal under the influence of trauma and infections. Therefore, various therapeutic strategies have been developed to promote bone regeneration in clinical settings. Autologous bone grafting is not subject to immune rejection and is the gold standard for repairing bone defects. However, there still remain some disadvantages such as limited donor sources and risk of tissue infection. Similarly, bone allografts, despite their wide variety of sources, carry risks such as immune rejection or disease transmission [1,2]. As a result, researchers have developed some implantable bioactive materials to repair bone defects, such as bioactive ceramics, polymers, and metal-based materials. These materials have great biocompatibility and high mechanical strength, which can be used to fill bone defects and cope with mechanical pulling in daily activities [3], and they do not have disadvantages such as autologous bone grafting or bone allografts. Conventional bone graft materials are mainly used to replace defective bone tissue, and they have limited effect on repairing bone defects. Therefore, an ideal bone graft material should promote osteogenesis and replace bone defects at the same time. Since metal elements play an important role in promoting bone healing in the process of bone regeneration, such as strontium (Sr), iron (Fe), magnesium (Mg), calcium (Ca), and tantalum (Ta), researchers have started to explore the application of metal elements in combination with bone graft materials in order to improve the bone regeneration and repair ability of conventional bone graft materials [4,5,6,7]. Compared with other ions, Sr has attracted the attention of researchers because of its dual regulatory ability to promote osteoblast proliferation and inhibit osteoclast activity [8,9]. Based on the development of technologies and the advent of therapeutic methods such as bone tissue engineering scaffolds, 3D printing, nanotechnology, and nano-drug delivery systems, it is possible to load Sr into biological transplantation materials [10,11].

Sr is a widespread trace element on earth, and it is present in the human body at low concentrations of about 10.57–12.23 mg/L. Sr is mostly deposited in the femur, lumbar spine, and iliac crest, with small amounts in the extracellular fluid [12]. Sr and Ca are 2-valent metal positive ions of the same family. Both of them are essential trace elements in the human body. Ca can induce the growth of bone precursor cells, stimulate the synthesis of osteoblasts, prolong the life of osteoblasts, and regulate the formation and reabsorption of osteoclasts in the process of bone regeneration [13]. Although the osteogenic effect of Sr in the human body has not been fully recognized, Sr and Ca have similar chemical properties, so their biological functions are also similar. Under physiological conditions, Sr has osteo-seeking properties and osteogenic effects similar to Ca [14]. However, excess Sr interferes with Ca metabolism and may lead to adverse consequences such as osteoporosis and skeletal deformities. Biomaterials can effectively control the release concentration of Sr, and several in vivo and in vitro studies have shown that the Sr compound with conventional bone graft biomaterials can further accelerate the repair of bone defects and enhance bone regeneration [15,16,17]. The purpose of this paper is to highlight the recent application of Sr in bone defect repair, and to provide a reference for its realization of clinical translation.

## 2. Mechanisms of Sr on Bone Regeneration

Sr plays an important part in the process of bone regeneration, but the relevant mechanisms have not been fully clarified. According to a study, Sr can regulate the differentiation of macrophages and modulate the local inflammatory response to promote early osteogenesis [18]. Another study found that Sr can promote the directional osteogenic differentiation of MSCs [19]. Others believe that Sr plays a role in promoting the proliferation and differentiation of osteoblasts and in inhibiting the activity of osteoclasts [20]. In this section, the relevant mechanisms of Sr in the process of bone regeneration will be briefly summarized (Figure 1).

### 2.1. Inflammatory Microenvironment

An appropriate immune microenvironment is crucial for the repair of bone defects, particularly favoring early vascular and bone tissue formation [5,21,22]. Biomaterials can modify the microenvironment at the implantation site by affecting the inflammatory response, which can induce the repair of bone defects [23]. Immune cells will be recruited to the surface of the biomaterials that are implanted in the body. Then, the physicochemical properties of the biomaterials can further induce immune responses and local tissue inflammation [24,25]. Subsequently, monocytes in the innate immune system differentiate into macrophages, and macrophages are stimulated by the local microenvironment to differentiate into pro-inflammatory macrophage type M1 and anti-inflammatory macrophage type M2, where type M1 is associated with energy metabolism and type M2 with tissue remodeling, repair, and wound healing [26]. Macrophages contact other immune cells by secreting bioactive factors to trigger immune responses and new bone formation [27]. Macrophages dominate the inflammatory response and also determine the effectiveness of bone repair. Therefore, bioactive materials for bone defect repair (such as bioactive ceramics, polymers, metal-based materials, and so on) should be beneficial in modulating the local inflammatory response and creating a suitable inflammatory microenvironment that can induce bone regeneration.

Sr induces macrophages to differentiate toward the pro-regenerative type M2 instead of the pro-inflammatory type M1 [28]. An in vitro study showed that Sr inhibited the inflammatory response of macrophages, and further weakened the inhibitory effect of the inflammatory response to osteogenic differentiation of bone marrow mesenchymal stem cells (BMSCs) [29]. Wang et al. [30] proved that Sr promoted the expression levels of anti-inflammatory factors of macrophage type M2 and improved osteoblast proliferation and differentiation through paracrine signaling. Fenbo et al. [18] proved that Sr had a positive effect on regulating the bone immune response of macrophages by lowering the expression of pro-inflammatory factors and catabolic genes and increasing the expression of osteoblast cell factors. Sr can also raise the expression of vascular endothelial growth factor (VEGF) and angiopoietin-1 (ANG-1), which facilitates early vascular regeneration [31]. Zhao et al. [32] observed that early efficient vascularization in the center of Sr-composited bioactive microspheres significantly promoted the formation of new bone in vivo. Therefore, Sr has the ability to control the local inflammatory reaction at the bone defect, which is beneficial to bone regeneration.

### 2.2. Mesenchymal Stem Cells

MSCs can differentiate into adipocytes, chondrocytes, and other osteoblast-related cells, but how to achieve directionally osteogenic differentiation remains an urgent problem. Sr can activate the Wnt/β-linked protein signaling pathway to regulate the proliferation, differentiation, and mineralization of human BMSCs in vitro [33]. Sr also raises the gene and protein expression of integrin β1 to promote the spread of aging human BMSCs, and decrease the transcriptional peroxisome proliferator-activated receptor γ (PPARγ), signal transducer, and transcriptional activator1, to inhibit the adipose differentiation of MSCs [34]. Lourenço et al. [35] proved the ability of Sr to induce osteogenic differentiation of MSCs and to reduce the number and function of osteoclasts. Wang et al. [36] prepared Sr-composited calcium silicate ceramics, significantly improving the regeneration of cartilage and subcartilaginous bone, which demonstrated that Sr could enhance the osteogenic and chondrogenic differentiation of MSCs. Zhou et al. [37] observed that the bone area, bone-implant contact, and removal torque values of Sr-composited implants were increased at the implantation site. Not only this, but Sr also promotes the proliferation of MSCs and maintains the cell population for osteogenic differentiation. Li et al. [38] found that Sr maintained more cell numbers in the cell cycle by raising the population of S and G2/M phase cells in initiating osteogenic differentiation, and the increased number of cells contributed to enhanced osteogenic differentiation. Cheng et al. [39] demonstrated that Sr-containing scaffolds were able to induce the osteogenic differentiation of MSCs. All of the above experiments are able to verify that Sr can promote the osteogenesis of MSCs in vivo and in vitro.

### 2.3. Osteoblasts

Bone morphogenetic protein-2 (BMP-2) is a member of the transforming growth factor-β (TGF-β) family, which can regulate osteogenic differentiation by inducing or promoting Runt-related transcription factor 2 (RUNX2) expressions [40]. BMP-2 can also promote the osteogenic differentiation of MSCs and early bone formation [41]. These proteins play a role in various developmental processes including bone formation, and cell proliferation and differentiation [40]. Sr is able to increase the RUNX2 and osteocalcin (OCN) expression of precursor osteoblasts and bone sialoprotein (BSP) and OCN expression of mature osteoblasts [42]. Sr also promotes osteoblast proliferation viability and stimulates the secretion of a new bone matrix, and Sr can inhibit osteoclast formation and differentiation [43]. Xie et al. [44] found that the dual regulatory properties of Sr are correlated with Ca concentration. At low Ca concentrations, Sr might inhibit the function of osteoblasts by decreasing alkaline phosphatase (ALP) activity and inhibiting the absorption of the bone-bridging protein and OCN. While at high Ca concentrations, Sr might enhance the effect of bone regeneration. RUNX2, BSP, Collagen I (COLI), and OCN are important factors in the process of osteogenesis, Sr was found to raise mRNA expression of these factors and increase ALP activity [45], and further study has confirmed these ideas [46]. In addition, osteoblasts can control bone absorption by regulating the formation of bone fractures through the osteoprotegerin (OPG)/receptor activator of the nuclear factor-κB ligand (RANKL) pathway [47].

### 2.4. Osteoclasts

Osteoclasts can absorb bone, and when bone homeostasis is out of balance, high osteoclast activity can cause bone absorption to exceed its formation, leading to the failure of bone defect repair. RANKL is a protein produced by osteoblast precursors, osteoblasts, and osteocytes, and RANK receptors are present on the surface of osteoclasts as well as on osteoblast precursors. OPG is synthesized by osteoblasts, and can act as a bait ligand for RNAK and reduce the binding of RNAKL to RANK receptors [48]. The RANKL signaling pathway plays a key role in the regulation of osteoclast formation, and Sr significantly inhibits the RANKL-induced p38 and NF-κB pathway. It ultimately leads to a reduction in the formation of osteoclasts and a lower expression of osteoclast-related genes [49]. Boanini et al. [50] cultured osteoclasts on Sr-substituted hydroxyapatite coating, and observed that the ratio of OPG/RANKL increased, which could prove the ability of the Sr-substituted coating to inhibit osteoclast formation and differentiation. Moseke et al. [51] prepared Sr-composited struvite coatings, and these coatings showed the inhibitory effect of Sr on osteoclasts at the morphology, biochemical, and gene expression level results.

### 2.5. Ca-sensitive Receptors

Ca-sensitive receptor (CaSR) is a G protein-coupled receptor, which plays a crucial role in regulating Ca concentration in the extracellular fluid and maintaining bone homeostasis. It senses extracellular fluid Ca concentration through cells in the parathyroid glands and renal tubules, and regulates parathyroid hormone secretion and renal Ca excretion to maintain proper Ca homeostasis in vivo [14]. CaSR exists in osteoblasts, osteoclast precursors, and mature osteoclasts. It regulates Ca concentration and the formation of bone tissue [52]. In addition to Ca, other divalent positive ions also activate CaSR, and Sr is a great CaSR agonist. Although its efficacy is slightly lower than that of natural Ca, it can greatly simulate the structure and overall properties of the Ca-binding site of CaSR, which offers the possibility that Sr promotes the repair of bone defects.

## 3. Biomaterials Compound with Sr

The development of suitable bone defect repair materials has been of importance to clinicians and researchers for a long time. With the development of bone biomaterials, bioactive ceramics, polymers, and metal-based materials have become good bone substitutes. However, simply implanting the materials into the body for bone defect repair is difficult to improve the long-term prognosis of patients. The correlation between trace elements in natural bone tissue and osteogenesis has been demonstrated; therefore, some scholars have proposed that incorporating trace elements into biomaterials to promote bone healing [53]. In this section, some Sr-composited biomaterials that promote bone regeneration will be briefly introduced (Figure 2).

### 3.1. Bioactive Ceramics

#### 3.1.1. Hydroxyapatite Scaffolds

Hydroxyapatite (HA) and its ion-substituted derivatives represent a large group of core inorganic materials for bone tissue engineering, and HA-based scaffolds are considered to be better materials for bone defect repair because of their chemical similarity to human bone and great biocompatibility. Sr-substituted HA enhances cell adhesion, cell proliferation, and ALP activity; the scaffold increases the osteogenic capacity of the body [54], and the implanted Sr-containing scaffold significantly increases the expression levels of osteogenic and angiogenic markers [55]. In addition, Sr-substituted apatite coating inhibits osteoclast activity, improves new bone formation, and enhances integration between bone and implants [56]. Chang et al. [57] developed a composite material by mixing Sr-substituted Ca sulfate hemihydrate (Sr-CSH) with HA, and it was implanted into a left shin bone defect in rats. Histological analysis showed that a large number of chondrocytes and osteoblasts had formed. The number of BMSCs, the expression of osteoblast marker genes, cell migration, and the area of mineralized nodules increased in Sr-CSH/HA. Ramadas et al. [58] prepared a Sr-substituted HA scaffold (Sr-HAP). In vivo tests showed that Sr-HAP successfully healed a 4 mm shin bone defect in rabbits after implantation in 45 days, and histological images showed it improved cell proliferation and new bone formation in the porous scaffold-treated group. Zhao et al. [59] prepared HA scaffolds, Sr-composited HA scaffolds, and HA scaffolds with the concomitant administration of SrRan. After 1 week of implantation, it was found that Sr-composited HA scaffolds or the administration of SrRan better induced the formation of vascular-like structures, but after 12 weeks of implantation, the Sr-composited HA scaffolds induced more new bone formation, and it had a lower blood Sr concentration and fewer adverse effects than the SrRan group. Moreover, Sr can stimulate macrophages to induce an immune microenvironment favorable to osteogenesis, and in vitro experiments have proved a promotional effect on the osteogenic differentiation of BMSCs [60]. Jiang et al. [61] prepared HA bioactive ceramics that were composited with different levels of Sr on the surface. In vitro evaluations showed that the bioactive ceramics could promote BMSC spread and proliferation, enhance ALP activity, and increase the gene expression of osteogenic and angiogenic factors, such as COLI, BSP, BMP-2, osteopontin (OPN), VEGF, and ANG-1. Meanwhile, HA ceramics composited with 10% Sr had the best stimulatory effect on promoting more bone and angiogenesis. Ge et al. [62] prepared the Sr-composited HA porous poly (l-lactic acid) scaffold. It can improve the hydrophobicity of the scaffold surface, reduce the degradation of the acidic environment, and enhance bone induction, and it had a good effect in promoting cell adhesion, proliferation, and ALP activity. Therefore, Sr-composited HA scaffolds can be used as a suitable bone defect repair material, which can support cell growth and proliferation because of their high compressive strength [63], high mineral adsorption rate [64], and strong bone integration ability [65] (Table 1).

#### 3.1.2. Bioactive Glass

Bioactive glass (BGs) is an inorganic biomaterial with high biocompatibility and bioactivity, mainly composed of silicon, Ca, and phosphorus oxides. These ions play an important role in cell proliferation as well as in homeostasis and bone remodeling, and the addition of small amounts of elemental oxides can confer osteogenic, angiogenic, antibacterial, anti-inflammatory, hemostatic, and anticancer traits on BGs [66]. Additionally, BGs in combination with organic substances can act as a template for cell adhesion, proliferation, and bone growth [67]. BGs can regulate and control immune cell responses to promote tissue regeneration [68]. Sr-substituted bioactive glass (Sr-BGs) is found to inhibit the formation of osteoclasts mediated by RANKL and lower the expression of osteoclast-related genes [49]. Sr-BGs also enhance ALP activity and Ca deposition, and increase collagen type I alpha 1 (ColIa1) and OCN expression [69]. Furthermore, an in vitro study researched by Baheiraei et al. [70] showed that gelatin-BGs/Sr scaffolds can inhibit the vitality of Escherichia coli and some Staphylococcus aureus, thereby preventing infection and improving bone regeneration. Sr-BGs with high mechanical strength and better cell differentiation efficiency are a suitable choice for bone defect repair materials [71]. Fiorilli et al. [72] confirmed the osteogenic effect of Sr incorporation into BGs by analyzing the expression of COLⅠa1, RANKL, OPG, and ALP. Moreover, Sr-BGs still have the ability to release Sr after biofunctionalization, and the down regulation of osteoclast differentiation genes also demonstrates the ability of Sr-BGs to inhibit osteoclast differentiation and function [73]. Wu et al. [74] prepared Sr-BGs, which have great biocompatibility in vivo and in vitro. They raised osteogenic and angiogenic abilities by activating the cyclic adenosine monophosphate/protein kinase A signaling pathway. Sr-BGs decreased the level of active oxygen in BMSCs in an osteoporosis model; thus, Sr-BGs could prevent osteoporosis during osteogenesis. In addition, Autefage et al. [75] designed a porous, Sr-releasing, bioactive glass-based scaffold (pSrBGs), and histological and morphological analyses showed that pSrBGs fitted tightly to bone tissue, greatly promoted lamellar bone formation, and the repaired new bone was similar to normal and healthy bone tissue. Shaltooki et al. [76] prepared polycaprolactone and Sr-containing BGs; this achieved good results in vitro experiments, including degradation tests, bioactivity tests, cytotoxicity tests, ALP activity tests, and cell adhesion tests. Furthermore, BGs also support the adhesion, colonization, and bone differentiation of BMSCs [19,77]. Sr-BGs prepared by Midha et al. [78] exhibited a superior ability to promote the osteogenic differentiation of BMSCs, such as toward osteoblasts and osteocytes. Fernandes et al. [79] developed Mg and Sr-substituted borate bioactive glass (BGs-Mg, BGs-Sr), both of them enhanced the expression of bone-specific proteins (ALP, OPG, and OCN), and the high mineralization of BMSCs under osteogenic medium conditions, but BGs-Sr was also able to increase the expression and mineralization of the same bone-specific proteins under basal medium conditions. This shows that Sr has a great effect on bone formation.

As is known to all, the BGs can be produced by both traditional melt-derived routes and sol–gel processes. BGs produced by high-temperature melting have limited activity because of their sharp decrease in hydroxyapatite formation ability, but sol–gel technology avoids this shortcoming and exhibits high biological activity potential [80]. In addition, the Sr-composited BGs made by the melting deposition method have a certain influence on the release of Sr [81], while the Sr-composited bioactive glass made by sol–gel exhibits high bone induction activity [82]. Based on the positive effects of Sr on osteogenesis, the development of Sr-composited silicate, borate, and phosphate-based BGs for bone defect repair is an advanced therapeutic strategy. Although it is controversial, the biological improvement of Sr-BGs on bone remodeling in vivo and in vitro is substantial and positive (Table 2).

#### 3.1.3. Ca Phosphate Ceramics

Ca phosphate has been widely used for bone regeneration due to its high biocompatibility and similarity to the human skeleton [83]. Ca and phosphorus can regulate osteoblast and osteoclast activation to promote osteogenesis, and the surface properties of Ca phosphate will influence adhesion and the growth of the cell and protein [84]. Ca is the basic element of bone, and the chemical properties of Sr and Ca are similar, so Sr can act through CaSR in bone tissue, and Sr-composited Ca phosphate ceramics have positive effects on bone defect repair [85]. Sr-containing Ca phosphate ceramics can enhance BMSC attachment and proliferation and significantly promote new bone regeneration in a rat bone defect model [86]. It promotes osteogenesis by raising the Wnt/β-linked signaling pathway; meanwhile, it inhibits osteoclast formation by lowering the NF-κB signaling pathway [87]. Tohidnezhad et al. [88] implanted Sr-composited β-tricalcium phosphate scaffolds into mice’s femoral defects for a period of 2 months, and it showed that Sr accelerated the bridging of the fracture gap. Tao et al. [89] investigated whether the topical administration of Sr and aspirin (Asp) could enhance tricalcium phosphate (β-TCP) in the treatment of osteoporotic bone defects. It showed that the cell mineralization degree and vitality of the Asp-Sr/β-TCP group were significantly increased, and the expressions of osteogenic proteins such as ALP, OPN, RUNX2, OCN, and COL1 were significantly increased. The imaging and histological results of the Asp-Sr/β-TCP group showed that bone regeneration and bone mineralization had the strongest effect. In addition, Sr-substituted Ca silicate ceramics have been shown to have a superior ability in promoting angiogenesis [90], and have promoted scaffold degradation and new bone maturation in a sheep shin bone defect model [91].

Bone cement can be used for bone defect repair due to its large surface area and strong protein loading capacity. Wu et al. [31] developed Sr-enhanced Ca phosphate hybrid cement. The incorporation of Sr enhanced the compressive strength of the cement, improved biocompatibility, increased ALP activity, Ca nodule formation, and related osteogenic gene expression, and Sr also raised the expression of VEGF and ANG-1. Reitmaier et al. [92] evaluated the short-term and long-term in vivo performance of Sr (II) calcium phosphate cement (SrCPC) scaffolds and CPC scaffolds. After implantation in sheep’s bone defects for 4 weeks, both scaffolds were penetrated by newly formed bone, and SrCPC did not significantly affect early osteogenesis. However, after 6 weeks, both SrCPC and CPC scaffolds showed good biocompatibility and bone binding capacity. However, the bone formation from SrCPC was more significant after 6 months. In addition, it has been demonstrated that Sr-composited bone cement helps to increase the proliferative activity of BMSCs [90], and the suitability of Sr-contained bone cement has been demonstrated in human cadaveric spine surgery [91] (Table 3).

#### 3.1.4. Other Bioactive Ceramics

There are also some ceramics with positive applications in bone defect repair. Zhang et al. [93] developed true bone ceramics combined with rhBMP-2 and Sr for bone induction and defect repair, and the results showed that the Sr-containing ceramics had significantly higher ALP activity, induced a small amount of new bone production, increased bone inductive activity, and it had the highest area of bone defect repair. Mao et al. [94] prepared bioactive ceramics containing Sr and silicon and found that these materials could enhance the ALP activity and expression of COL1, OCN, RUNX2, and angiogenic factors (including VEGF and Ang-1). Meanwhile, Sr and silicon had synergistic effects on osteogenesis, osteoclastogenesis, and angiogenesis.

### 3.2. Polymers

Both natural and synthetic polymers are constantly being studied and applied in the biomedical field. In order to suit the human body’s needs, they can be manufactured and synthesized with artificially controlled parameters such as biocompatibility and mechanical strength. So, they will become good substitutes for bone defect repair [95].

#### 3.2.1. Natural Polymers

Ye et al. [96] developed Sr-composited Ca phosphate/polycaprolactone/chitosan (Sr-CaP/PCL/CS) nanohybridization membranes, which mimicked the extracellular matrix structure while constantly allowing the release of Sr to promote bone regeneration. In vitro cell culture demonstrated that the membranes significantly promoted adhesion and proliferation of rat’s BMSCs. Moreover, it exhibited higher ALP activity and higher matrix mineralization in terms of osteogenic differentiation. More importantly, the synergistic effect of Sr enhanced the angiogenic differentiation of BMSCs. Ma et al. [97] found that a novel polysaccharide–metal complex Sr Laminarin polysaccharide (LP-Sr) effectively promoted VEGF and epidermal growth factor-like domain multiple six expressions, and significantly raised ColIa1 and OCN expression. LP-Sr had a positive inhibitory effect on the pro-inflammatory factor interleukin-6, and the markers of osteogenic and angiogenic (ALP and CD31) were highly expressed. Wu et al. [98] developed a biodegradable serine protein-gelatin scaffold doped with SrP and ginsenoside Rg1. This scaffold stimulated the osteogenic differentiation of mouse BMSCs and promoted human umbilical vein endothelial cell angiogenesis by activating the expression of vascular endothelial growth factor and basic fibroblast growth factor genes and proteins. In addition, the scaffold-released Sr and Rg1 also lowered the expression of inflammation-related genes, and results in vivo showed that the scaffold significantly promoted bone repair in a model of osteoporotic skull defects. Luo et al. [99] developed a Sr-Ca sulfate hemihydrate scaffold incorporating a ginsenoside Rg1/gelatin microsphere. It promoted bone tissue repair and regeneration in vivo and had a good ability to promote osteogenic differentiation and angiogenesis in vitro. Cheng et al. [39] coated SrCl on a surface porous calcined porcine bone scaffold containing polycaprolactone (CPB/PCL/Sr), PCL was able to improve the mechanical properties of the scaffold and inhibit the release of Sr, and CPB/PCL/Sr supported the osteogenic differentiation of MSCs better than CPB. Xu et al. [100] designed a metformin hydrochloride encapsulated Sr alginate hydrogel (Alg/MH-Sr). RT-PCR tests showed that Alg/MH-Sr significantly inhibited senescence, apoptosis, oxidative, and inflammatory gene expression, and increased chondrocyte repair. Repairing chondrocytes may be an effective application direction for bone defect repair. Xu et al. [101] developed a chitosan-Sr sulfate chondroitin scaffold, it was able to lower the expression of inflammatory and osteoclast-related mRNA while increasing BMP-2 expression, and this scaffold promoted bone defect healing in an aged rat bone defect model. Hassani et al. [102] incorporated Ca, Ba, and Sr alginate-nanohydroxyapatite-collagen microspheres. Sr-containing microspheres were able to enhance the viability of human MG-63 osteoblasts and osteogenic capacity. CT and histological examination analysis showed that Sr-containing microspheres promoted the healing of skull defects and accelerated bone formation in rats (Table 4).

#### 3.2.2. Synthetic Polymers

Lourenço et al. [35] designed a Sr cross-linked arginine-glycine-aspartic acid-alginate hydrogel enhanced with HA microspheres, and in vitro tests confirmed its ability to induce the osteogenic differentiation of MSCs and to reduce osteoclast function. The hydrogel was implanted into an in vivo inflammation model, and it was able to modulate the inflammatory response through macrophage-type M2 polarization. Gao et al. [103] synthesized novel Sr-HA-graft-poly (γ-benzyl-l-glutamate) nanocomposite microcarriers, which promoted cell adhesion, proliferation, and increased extracellular matrix secretion. Meanwhile, it effectively promoted osteogenic gene expression, and this material could promote bone regeneration at non-healing sites after 8 weeks of implantation in a mouse model. Lino et al. [104] developed a compatible blend of poly-ε-caprolactone (PCL) and polydiisopropyl fumarate (PDIPF) enriched with 1% or 5% Sr. In vitro, this polymer released very low levels of positive ions and did not have cytotoxicity to cultured macrophages. In vivo, implants containing 1% Sr significantly increased bone tissue regeneration and improved fibrous bridging without inducing local inflammatory responses or increasing serum Sr levels. Han et al. [105] prepared mineralized electrostatic spun poly (lactic acid) nanofiber membranes containing varying amounts of Sr, and the membrane promoted BMSC proliferation and osteogenic differentiation, and in vivo bone defect experiments also proved that the membrane could promote bone regeneration. Lin et al. [106] developed a Sr peroxide (SrO_2_)-loaded poly (lactic-co-glycolic acid) (PLGA)-gelatin scaffold system. This system effectively stimulated osteoblast proliferation and inhibited osteoclast formation. Ray et al. [107] prepared Sr and bisphosphonate-coated iron foam scaffolds (FeSr) for osteoporotic fracture defect healing. In a rat model, bone formation at the interface of FeSr implantation increased, accompanied by an increase in osteoblasts and a decrease in osteoclast activity, and immunohistochemical results showed that BMP-2 and RNAKL/OPG decreased (Table 5).

### 3.3. Metal-Based Materials

Bioactive ceramics and polymers have been extensively studied as bone defect repair materials, but there is still much room for improvement in their mechanical properties such as mechanical strength and fatigue resistance, which can be well addressed by metal-based materials. Titanium (Ti) has good chemical and mechanical stability, biocompatible, mechanical strength, and corrosion resistance. All kinds of bio-functional molecular immobilization technologies that have been developed can be used for bone formation and the prevention of platelet and bacterial adhesion, which makes it possible for Sr to be applied to Ti-based materials [108]. Ti-Sr bound nanotubes can effectively inhibit osteoclast differentiation by inhibiting NF-κB and Akt/NFATc1 pathways, as well as negatively regulating the ERK pathway in vivo and in vitro [109]. Sr-composited Ti implants accelerated bone healing [110] and significantly raised macrophage phenotype and anti-inflammatory factor production to enhance bone integration [111]. Ding et al. [112] prepared protein supramolecular nanofilm (Ti-Ly-Sr) composited with Sr on Ti, cell morphology observation, cell activity assay, ALP staining, and quantitative analysis showed that Ti-Ly-Sr enhanced the early adhesion, proliferation, and osteogenic differentiation of BMSCs, increased the expression of BMSC-related osteogenic genes such as BMP-2, OPG, Runx2, and COL1. In addition, Ti-Ly-Sr promoted new bone formation after implantation in 4 weeks. Xu et al. [113] evaluated the effect of Sr-Ti implants on bone integration in diabetic rats. The implant lowered the expression of relevant inflammatory factors, such as tumor necrosis factor-α, interleukin-1β (IL-1β), and IL-6, after implantation in 3 days, with the expression of OPG being raised after 4 weeks, and the percentage of implant contact significantly higher after 4 and 8 weeks. By evaluating Sr-composited titanium dioxide coatings, Zhou et al. [114] found that Sr improved MSC proliferation, osteogenic differentiation, and bone integration of the implant; meanwhile, the angiogenesis and antibacterial ability of the coating was not weakened by Sr. Li et al. [115] designed a dual delivery system coated on the Ti surface. This system could manipulate macrophage polarization to activate osteoblast pre-differentiation. Li et al. [116] found that Sr-composited titanium dioxide mesoporous nanospheres greatly promoted the formation of new bone in extraction sockets. In addition, Sr-containing implants had a positive effect on the early bone integration effect in osteoporotic rabbits [117]. Furthermore, Ta-Sr [118], Zn-Sr [119], Mg-Sr [120], and Mg-Ti [121] alloys could also better promote bone formation and mineralization, and there was a synergistic antibacterial behavior between Sr and silver [122]. Compared with other ions, Ta is an element with high chemical stability and ductility that can be used in orthopedic biomaterials. The application of Ta-Sr material to orthopedic implants has the latent capacity to increase the lifetime of the implants [118]. Therefore, a Sr-dropped Ta metal-based material may have great potential in future clinical applications. The above instances indicate that Sr has broad application prospects in metal-based materials (Table 6).

## 4. Conclusions and Perspectives

The Sr compound with biomaterials is a promising therapeutic strategy for bone tissue regeneration, and it has shown better repair effects than conventional biomaterials in multiple studies. However, an ideal bone defect repair material should not only have good biocompatibility and degradability, but should also have the ability to control the release concentration of Sr in vivo. This can reduce the adverse effects on other systems of the body. In addition, the combination of Sr and biomaterials will affect the repair effect of Sr, which also limits the application of the Sr compound with biomaterials to a certain extent. The above two points are the problems that hinder the clinical application of Sr combined with biomaterials. In the future, we may devote ourselves to improving the ability of biomaterials to release Sr and reducing the influence of biomaterials on Sr in vivo. This will enable us to realize the clinical translation of Sr compound with biomaterials as soon as possible.

The Sr compound with biomaterials has the advantages of easy acquisition and convenient adjustment properties, and it has a wide range of clinical application prospects. The functionalization of biomaterials and Sr used in bone regeneration strategies has achieved good results in previous studies, where it has been shown to improve bone healing by enhancing local bone regeneration. With the deepening of research and the development of emerging technologies such as 3D printing and nanotechnology, the Sr compound with biomaterials will gradually meet the needs of patients and eventually achieve large-scale clinical applications.

## Figures and Tables

**Figure 1 bioengineering-10-00414-f001:**
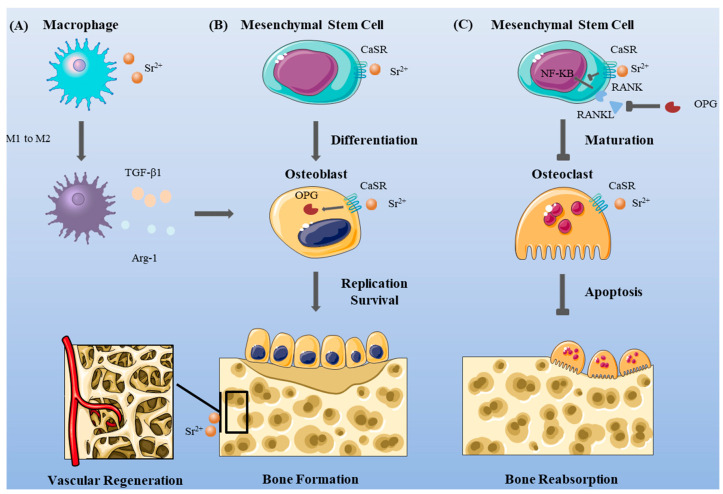
Mechanisms of Sr on bone regeneration: (**A**) Sr induces macrophages to differentiate toward M2 instead of M1, which benefits the promotion of osteoblast proliferation, and Sr also raises early vascular regeneration. (**B**) Sr promotes mesenchymal stem cell differentiation and osteoblast proliferation, which benefits bone formation. (**C**) Sr inhibits mesenchymal stem cell differentiation and osteoclast proliferation, which reduces bone reabsorption.

**Figure 2 bioengineering-10-00414-f002:**
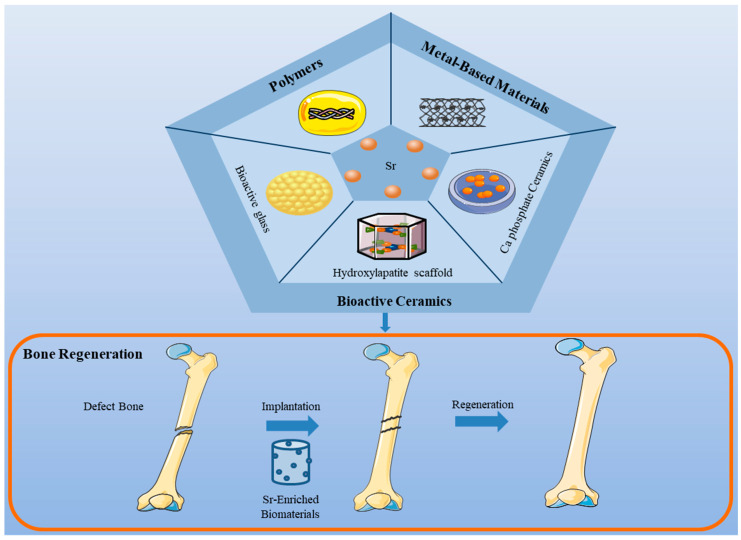
Biomaterials compound with Sr for bone regeneration.

**Table 1 bioengineering-10-00414-t001:** The application of Sr compound with hydroxyapatite scaffolds.

Year	Team	Materials	Results
2018	Luo et al. [54]	Sr-substituted HA scaffold	Increased adhesion, proliferation, and ALP activity of MC3T3-E1
2018	Ge et al. [62]	Sr-composited HA porous poly scaffold	Increased adhesion, proliferation, and ALP activity of MC3T3-E1
2019	Oryan et al. [55]	Incorporation of Sr and bioglass into G/nHAp scaffold	Increased expression of OPN, OCN, and angiogenic markers of BMSCs
2020	Geng et al. [56]	Nano-needle Sr-substituted apatite coating	Increased adhesion, spreading, proliferation, and osteogenic differentiation of BMSCs, inhibited differentiation of osteoclasts
2020	Chang et al. [57]	Sr-substituted calcium sulfate hemihydrate/HA scaffold	Increased proliferation, migration, mineralized nodule area, and differentiation into osteoblast-like cells of BMSCs
2020	Zhao et al. [59]	Sr-substituted HA scaffold	Increased expression of the osteogenic marker in BMSCs
2021	Ramadas et al. [58]	Sr-substituted HA scaffold	Increased proliferation of MG-63
2022	Zhong et al. [60]	Zn/Sr dual ion-collagen co-assembly HA	Increased osteogenic differentiation of BMSCs
2022	Jiang et al. [61]	Bioactivity of HA doped with different levels of Sr ceramics	Increased the proliferation, ALP activity, and gene expression of osteogenic and angiogenic factors in BMSCs

ALP: alkaline phosphatase, BMSCs: bone marrow mesenchymal stem cells, G/nHAp: gelatin/nano-hydroxyapatite, HA: hydroxyapatite, OPN: osteopontin, OCN: osteocalcin.

**Table 2 bioengineering-10-00414-t002:** The application of Sr compound with bioactive glass.

Year	Team	Materials	Results
2017	Fernandes et al. [79]	Mg and Sr-substituted BGs	Increased osteogenic differentiation and expression of ALP, OPN, and OCN in BMSCs
2018	Naruphontjirakul et al. [69]	Sr-containing BG nanoparticles	Increased ALP activity and expression of OCN in MC3T3-E1
2018	Fiorilli et al. [72]	Sr-BGs	Increased osteogenic differentiation of SAOS-2
2018	Midha et al. [78]	Sr-BGs	Increased osteogenic differentiation of BMSCs
2019	Autefage et al. [75]	PSrBG	Increased proliferation of BMSCs and MC3T3-E1
2019	Shaltooki et al. [76]	BGs composed of PCL and different levels of Sr	Increased osteogenic activity of MG-63
2020	Huang et al. [49]	Sr-substituted BGs	Inhibited RANKL-mediated osteoclastogenesis
2021	Baheiraei et al. [70]	Gel-BG/Sr scaffolds	Increased bone formation
2021	Fiorilli et al. [73]	Sr-containing esoporous BGs	Inhibited osteoclast differentiation and function
2022	Wu et al. [74]	Sr-BG	Increased osteogenesis and angiogenesis of BMSCs

ALP: alkaline phosphatase, BMSCs: bone marrow mesenchymal stem cells, BGs: bioactive glasses, Gel-BG: gelatin-bioactive glasses, OPN: osteopontin, OCN: osteocalcin, PCL: polycaprolactone, PSrBG: porous, Sr-releasing and bioactive glass-based scaffold, RANKL: receptor activator of nuclear factor-κB ligand.

**Table 3 bioengineering-10-00414-t003:** The application of Sr compound with calcium phosphates ceramics and bone cements.

Year	Team	Materials	Results
2018	Reitmaier et al. [92]	Sr(II)-doted CPC scaffolds	Increased bone formation
2019	Li et al. [91]	Sr-hardystonite-gahnite bioactive ceramic scaffold	Induced substantial bone formation and defect bridging
2020	Chen et al. [86]	Sr-substituted biphasic calcium phosphate microspheres	Increased proliferation and osteogenic inductivity of BMSCs
2020	Zeng et al. [87]	Sr-substituted calcium phosphate silicate bioactive ceramic	Increased proliferation and ALP activity of BMSCs, inhibited osteoclast differentiation
2020	Tohidnezhad et al. [88]	Sr-composited β-tricalcium phosphate scaffold	Increased bone fracture gap bridging
2020	Tao et al. [89]	Aspirin-modified Sr-composited β-tricalcium phosphate	Increased osteogenic viability of MC3T3-E1
2020	Wu et al. [31]	Sr-reinforced calcium phosphate hybrid cement	Increased ALP activity and osteogenic gene expression of BMSCs, and promoted bone regeneration
2021	Liu et al. [90]	Sr-substituted calcium silicate ceramics	Increased angiogenesis of BMSCs and accelerated bone regeneration

ALP: alkaline phosphatase, BMSCs: bone marrow mesenchymal stem cells, CPC: calcium phosphate cements.

**Table 4 bioengineering-10-00414-t004:** The application of Sr compound with natural polymers.

Year	Team	Materials	Results
2018	Cheng et al. [39]	SrCl-coated surface porous CPB scaffold containing PCL	Increased osteogenic differentiation of BMSCs
2019	Ye et al. [96]	Sr-composited calcium phosphate/polycaprolactone/chitosan nanohybrid films	Increased adhesion, proliferation, and vascular differentiation of BMSCs
2020	Luo et al. [99]	Sr-calcium sulfate hemihydrate scaffold containing ginsenoside Rg1-encapsulated gelatin microspheres	Increased osteogenic differentiation and ALP activity of MC3T3-E1
2021	Ma et al. [97]	Sr Laminarin polysaccharide	Increased expression of OCN in MC3T3-E1
2021	Wu et al. [98]	Biodegradable silk protein-gelatin scaffolds doped with SrP and ginsenoside Rg1	Increased osteogenic differentiation of BMSCs
2021	Xu et al. [100]	Metformin hydrochloride encapsulated Sralginate hydrogel	Increased chondrocyte repair, inhibited expression of senescence apoptosis, oxidative, and inflammatory genes
2021	Xu et al. [101]	Chitosan-Sr sulfate chondroitin scaffold	Increased BMP-2 expression of MC3T3-E1
2022	Hassani et al. [102]	Alginate-nano-hydroxyapatite-collagen microspheres mixed with Ca^2+^, Ba^2+^, and Sr^2+^	Increased the viability and osteogenic capacity of osteoblasts

ALP: alkaline phosphatase, BMSCs: bone marrow mesenchymal stem cells, BMP-2: bone morphogenetic protein-2, CPB: calcined porcine bone, OCN: osteocalcin, PCL: polycaprolactone.

**Table 5 bioengineering-10-00414-t005:** The application of Sr compound with synthetic polymers.

Year	Team	Materials	Results
2017	Gao et al. [103]	Sr-HA-graft-Poly (γ-benzyl-l-glutamate) nanocomposite microcarriers	Increased adhesion, proliferation, and osteogenic gene expression of ADSCs
2019	Lourenço et al. [35]	Sr-crosslinked RGD-alginate hydrogel reinforced with Sr-doped hydroxyapatite microspheres	Induced osteogenic differentiation of BMSCs and reduced osteoclast function
2019	Lino et al. [104]	A compatibilized blend of poly-ε-caprolactone and polydiisopropyl fumarate enriched with 1% or 5% Sr^2+^	Increased expression of ALP in BMSCs
2019	Han et al. [105]	Mineralized electrostatic spun poly (lactic acid) nanofiber membranes with different amounts of Sr	Increased proliferation and osteogenic differentiation of BMSCs
2022	Lin et al. [106]	Sr peroxide-loaded poly (lactic-*co*-glycolic acid)-gelatin scaffold system	Increased proliferation of osteoblast and inhibited formation of osteoclast

ADSCs: adipose-derived stem cells, ALP: alkaline phosphatase, BMSCs: bone marrow mesenchymal stem cells, HA: hydroxyapatite, RGD: arginine-glycine-aspartic acid.

**Table 6 bioengineering-10-00414-t006:** The application of Sr compound with metal-based materials.

Year	Team	Materials	Results
2017	Mi et al. [109]	Sr-loaded Ti dioxide nanotube	Inhibited osteoclast differentiation
2018	Choi et al. [111]	Sandblasted/acid-etched titanium implants with Sr-containing nanostructures	Increased osteogenic differentiation of BMSCs and expression of osteogenic genes in osteoblasts
2019	Zhou et al. [114]	Sr-composited titanium dioxide coating	Increased proliferation and osteogenic differentiation of BMSCs
2019	Li et al. [115]	Dual delivery system coated on Ti surface	Manipulated macrophage polarization to activate pre-osteoblast differentiation
2019	Lin et al. [117]	Sr-incorporated titanium implant	Increased effect of early bone healing
2020	Ding et al. [112]	Protein supramolecular nanomembranes doped with Sr on Ti base	Increased early adhesion, proliferation, osteogenic differentiation, and expression of osteogenic genes in BMSCs
2020	Jia et al. [118]	Zn-Sr alloy	Increased cytocompatibility and osteogenesis of MC3T3-E1
2020	Zhang et al. [119]	Mg-Sr alloy	Increased proliferation, mineralization, and ALP activity of BMSCs
2021	Xu et al. [113]	Sr-Ti implants	Increased OPG expression and lowered inflammatory factors expression
2022	Su et al. [110]	Sr calcium phosphate coating on Ti6Al4V scaffolds	Increased adhesion, spreading, and osteogenesis of BMSCs
2022	Li et al. [116]	Sr-doped titanium dioxide mesoporous nanospheres	Increased the formation of new bone tissue

ALP: alkaline phosphatase, BMSCs: bone marrow mesenchymal stem cells, OPG: osteoprotegerin, Ti: titanium.

## Data Availability

No new data were created or analyzed in this study. Data sharing is not applicable to this article.

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
