# Peer review of "Recent Advance of Strontium Functionalized in Biomaterials for Bone Regeneration"

_bioengineering, 2023, doi:10.3390/bioengineering10040414_

Round 1
Reviewer 1 Report
The manuscript is a well-documented review based on recent bibliographic sources. It consists of two major parts: one related to strontium (Sr) mechanisms involved in bone regeneration and the second to biomaterials with Sr. The manuscript is very well organized, with accurate biomaterial classification, easy to read, providing sufficient information for a facile understanding of the topic.
I have the following suggestions:
- Please check your manuscript for minor English errors.
- Please consider extending the Conclusion and Perspectives section with a more ample discussion on the subject, with specific research gaps related to the use of strontium in biomaterials for bone regeneration and some possible research directions/research trends that may address those gaps.
Author Response
The manuscript is a well-documented review based on recent bibliographic sources. It consists of two major parts: one related to strontium (Sr) mechanisms involved in bone regeneration and the second to biomaterials with Sr. The manuscript is very well organized, with accurate biomaterial classification, easy to read, providing sufficient information for a facile understanding of the topic.
I have the following suggestions:
1.Please check your manuscript for minor English errors.
Response: We are grateful for this suggestion. We completely agree with this valuable suggestion by the reviewer. As suggested by the reviewer, we have corrected English errors in the revised manuscript.
2.Please consider extending the Conclusion and Perspectives section with a more ample discussion on the subject, with specific research gaps related to the use of strontium in biomaterials for bone regeneration and some possible research directions/research trends that may address those gaps.
Response: We completely agree with this valuable suggestion by the reviewer. As suggested by the reviewer, we have added some specific research gaps related to the use of strontium in biomaterials for bone regeneration, and supplemented some possible research directions that may address those gaps.
Details: However, an ideal bone defect repair material should not only have good biocompatibility and degradability, but also have the ability to control the release concentration of Sr in vivo. This can reduce the adverse effects on other systems of the body. In addition, the combination of Sr and biomaterials will affect the repair effect of Sr, which also limits the application of Sr compound with biomaterials to a certain extent. The above two points are the problems that hinder the clinical application of Sr combined with biomaterials. In the future, we may devote ourselves to improving the ability of biomaterials to release Sr and reducing the influence of biomaterials on Sr in vivo. This will enable us to realize the clinical translation of Sr compound with biomaterials as soon as possible.

Reviewer 2 Report
The review “Recent advance of strontium functionalized in biomaterials for bone regeneration” the latest studies of Sr combined with biomaterials, and the relevant mechanisms of Sr in the process of bone regeneration.
In the bioactive glasses section (3.1.2), a comparative remark between Sr doped melt derived and sol-gel glasses.
The Conclusions section needs more discussion. I suggest a summary comparing which Sr-doped biomaterials exhibit outstanding properties related to bone regeneration and which have potential or are used in clinical applications.
The references are not written following the Journal's Author's Instructions.
Ray S, Thormann U, Eichelroth M, et al., 2018, Strontium and bisphosphonate coated iron foam scaffolds for osteoporotic fracture defect healing [J]. Biomaterials, 157: 1-16
However, no cited work of the authors related to the review's subject can be found in the references section.
Author Response
Reviewer 2
The review “Recent advance of strontium functionalized in biomaterials for bone regeneration” the latest studies of Sr combined with biomaterials, and the relevant mechanisms of Sr in the process of bone regeneration.
1.In the bioactive glasses section (3.1.2), a comparative remark between Sr doped melt derived and sol-gel glasses.
Response: Thank you for your valuable comment. We completely agree with this valuable suggestion by the reviewer. As suggested by the reviewer, we have supplemented a comparative remark between Sr doped melt derived and sol-gel glasses.
Details:As is known to all, the BGs can be produced by both a traditional melt derived route and sol-gel process. BGs produced by high temperature melting have limited activity because of its sharp decrease in hydroxyapatite formation ability, but sol-gel technology avoids this shortcoming and exhibits high biological activity potential [1]. In addition, the Sr composited BGs made by melting deposition method have certain influence on the release of Sr [2], while the Sr composited bioactive glass made by sol-gel exhibit high bone induction activity [3].
Added references:
(1) Fiume, E.; Migneco, C.; Verné, E.;Baino, F. Comparison Between Bioactive Sol-Gel and Melt-Derived Glasses/Glass-Ceramics Based on the Multicomponent SiO(2)-P(2)O(5)-CaO-MgO-Na(2)O-K(2)O System [J]. Materials (Basel) 2020, 13(3).
(2) Brauer, D.S.; Karpukhina, N.; Kedia, G.; Bhat, A.; Law, R.V.; Radecka, I.;Hill, R.G. Bactericidal strontium-releasing injectable bone cements based on bioactive glasses [J]. J R Soc Interface 2013, 10(78), 20120647.
(3) Leite Á, J.; Gonçalves, A.I.; Rodrigues, M.T.; Gomes, M.E.;Mano, J.F. Strontium-Doped Bioactive Glass Nanoparticles in Osteogenic Commitment [J]. ACS Appl Mater Interfaces 2018, 10(27), 23311-23320.
2.The Conclusions section needs more discussion. I suggest a summary comparing which Sr-doped metal-based materials exhibit outstanding properties related to bone regeneration and it have potential or are used in clinical applications.
Response: We are grateful for this valuable suggestion. As suggested by the reviewer, we have supplemented Sr-dopped Ta materials exhibit outstanding properties related to bone regeneration and have great clinical application potential.
Details:Compared with other ions, Ta is an element with high chemical stability and ductility that can be used in orthopedic biomaterials. The application of Ta-Sr material to orthopedic implants has the latent capacity to increase the lifetime of the implants [1]. Therefore, Sr-dropped Ta metal-based material may have great potential in future clinical applications.
Added references:
(1) Li, R.; Wei, Y.; Gu, L.; Qin, Y.;Li, D. Sol-gel-assisted micro-arc oxidation synthesis and characterization of a hierarchically rough structured Ta-Sr coating for biomaterials [J]. RSC Adv 2020, 10(34), 20020-20027.
3.The references are not written following the Journal's Author's Instructions.
Ray S, Thormann U, Eichelroth M, et al., 2018, Strontium and bisphosphonate coated iron foam scaffolds for osteoporotic fracture defect healing [J]. Biomaterials, 157: 1-16
Response: We completely agree with this valuable suggestion by the reviewer. As suggested by the reviewer, the references have been corrected following the Journal's Author's Instructions
4.However, no cited work of the authors related to the review's subject can be found in the references section.
Response: We are grateful for this suggestion. I apology for a mistake in our work, after careful examination, we have deleted the section which not extremely relevant to the subject of this manuscript.

Reviewer 3 Report
The work titled „Recent advance of strontium functionalized in biomaterials for bone regeneration” is a summary of state of the art related to using strontium for bone regeneation in different types of materials like ceramics, glas, metalic materials and polymers, but work also included disscusion part describing bone regeneation mechanism. The summary is quite intersting hawever requires some minor corrections before publication.
line 49 Some examples of the metal elements should be presented. This part could include comparison of strontium with other elements with highlighting the advantages with particular elements.
line 93 Some examples of the materials (e.g. their groups) can be presented (even if they are described in next sections)
line 109 correct into „in vitro”
line 123 change into full name of the abbreviation
line 266 „in vitro and in vitro” – it should be correct
line 333 Change the title (more specific)
References: correct the style according to Journal requirements (e.g. Name with dots, Name of journal in intalics)
Author Response
Reviewer3
The work titled „Recent advance of strontium functionalized in biomaterials for bone regeneration” is a summary of state of the art related to using strontium for bone regeneation in different types of materials like ceramics, glas, metalic materials and polymers, but work also included disscusion part describing bone regeneation mechanism.
1.The summary is quite intersting hawever requires some minor corrections before publication.
Response: Thank you for your valuable comment. We completely agree with this valuable suggestion by the reviewer. As suggested by the reviewer, we have corrected the summary.
Details:Bone defect disease is an important reason which causes of people`s life and property damage, and how to effectively promote bone regeneration is still a big clinical challenge. Most of the current repair methods focus on filling the defects, which have poor effect on bone regeneration. Therefore, how to effectively promote bone regeneration and repair defects at the same time has become a challenge for clinicians and researchers. Strontium (Sr) is a trace element required by the human body, which mainly exists in human bones. Due to its unique dual property of promoting proliferation and differentiation of osteoblast and inhibiting osteoclast activity, it has attracted extensive researches of bone defect repair in recently years. With the deep development of researches, the mechanisms of Sr in the process of bone regenerationhave been clarified, and the effects of Sr on osteoblasts, osteoclasts, mesenchymal stem cells (MSCs) and inflammatory microenvironment have been widely recognized. Based on the development of technology such as bioengineering, it`s possible that Sr can be better loaded onto biomaterials. Despite the clinical application of Sr is currently limited and relevant clinical researches are still need to be developed, Sr-composited bone biomaterials have achieved satisfactory results in vitro and in vivo studies. Sr compound with biomaterials to promote bone regeneration will be a development direction in the future. This review will present a brief overview of the relevant mechanisms of Sr in the process of bone regeneration and the related latest studies of Sr combined with biomaterials. The aim of this paper is to highlight the potential prospects of Sr functionalized in biomaterials.
2.line 49 Some examples of the metal elements should be presented. This part could include comparison of strontium with other elements with highlighting the advantages with particular elements.
Response: We are grateful for this suggestion. As suggested by the reviewer, We added some metal elements such as Fe, Mg , Ca and Ta to this part of lines 51-58 to highlight the advantages of Sr.
Details:Since metal elements play an important role in promoting bone healing in the process of bone regeneration, such as Stromtium (Sr), ferrum (Fe), magnesium (Mg), calcium (Ca) and tantalum (Ta), researchers have started to explore the application of metal elements in combination with bone graft materials in order to improve the bone regeneration and repair ability of conventional bone graft materials [1-3]. Compared with other ions, Sr has attracted the attention to researchers because of its dual regulatory ability to promote osteoblast proliferation and inhibit osteoclast activity.
Added reference:
(1) Qiao, W.; Wong, K.H.M.; Shen, J.; Wang, W.; Wu, J.; Li, J.; Lin, Z.; Chen, Z.; Matinlinna, J.P.; Zheng, Y.; et al. TRPM7 kinase-mediated immunomodulation in macrophage plays a central role in magnesium ion-induced bone regeneration [J]. Nat Commun 2021, 12(1), 2885.
(2) Fischer, V.; Haffner-Luntzer, M.; Amling, M.;Ignatius, A. Calcium and vitamin D in bone fracture healing and post-traumatic bone turnover [J]. Eur Cell Mater 2018, 35, 365-385.
(3) Dommeti, V.K.; Roy, S.; Pramanik, S.; Merdji, A.; Ouldyerou, A.;Özcan, M. Design and Development of Tantalum and Strontium Ion Doped Hydroxyapatite Composite Coating on Titanium Substrate: Structural and Human Osteoblast-like Cell Viability Studies [J]. Materials (Basel) 2023, 16(4).
3.line 93 Some examples of the materials (e.g. their groups) can be presented (even if they are described in next sections)
Response: We completely agree with this valuable suggestion by the reviewer. As suggested by the reviewer, we have added bioactive ceramics, polymers and metal-based materials to this part of lines 99-102
Details:Therefore, bioactive materials for bone defect repair (such as bioactive ceramics, polymers, metal-based materials and so on) should be benefit to modulating the local inflammatory response and creating a suitable inflammatory microenvironment which can induce bone regeneration.
4.line 109 correct into „in vitro”
Response: We are grateful for this suggestion. As suggested by the reviewer, we have corrected “as vitro” to “in vitro” in line 104.
Details:In vitro study showed that Sr inhibited the inflammatory response of macrophages, and further weakened the inhibitory effect of the inflammatory response to osteogenic differentiation of bone marrow mesenchymal stem cells (BMSCs).
5.line 123 change into full name of the abbreviation
Response: We are grateful for this suggestion. As suggested by the reviewer, we have corrected “MSCs” to “Mesenchymal stem cells” in line 117.
Details:Mesenchymal stem cells
6.line 266 „in vitro and in vitro” – it should be correct
Response: We are grateful for this suggestion. As suggested by the reviewer, we have corrected “in vitro and in vitro” to “in vivo and in vitro” in line 245.
Details:Wu et al. prepared Sr-BGs, which had great biocompatibility in vivo and in vitro.
7.line 333 Change the title (more specific)
Response: We completely agree with this valuable suggestion by the reviewer. As suggested by the reviewer, we have changed the title to “other bioactive ceramics” in line 304, and we have deleted and corrected the content of the part, which can make the title more specific.
Details:Mao et al. [1] prepared bioactive ceramics contain Sr and silicon, and this materials could enhance the ALP activity and expression of COL1, OCN, Runx2 and angiogenic factors (including VEGF and Ang-1). Meanwhile, Sr and silicon had synergistic effects on osteogenesis, osteoclastgenesis and angiogenesis.
Added reference:
(1) Mao, L.; Xia, L.; Chang, J.; Liu, J.; Jiang, L.; Wu, C.;Fang, B. The synergistic effects of Sr and Si bioactive ions on osteogenesis, osteoclastogenesis and angiogenesis for osteoporotic bone regeneration [J]. Acta Biomater 2017, 61, 217-232.
8.References: correct the style according to Journal requirements (e.g. Name with dots, Name of journal in intalics)
Response: We are grateful for this suggestion. As suggested by the reviewer, We have re-examined and proofread all the references in this paper.

Round 2
Reviewer 2 Report
The manuscript is acceptable for publication in its present form